# Genome-wide CRISPR/Cas9 deletion screen defines mitochondrial gene essentiality and identifies routes for tumour cell viability in hypoxia

Luke W. Thomas [1,4], Cinzia Esposito[1,3,4], Rachel E. Morgan[1], Stacey Price[2], Jamie Young[2], Steven P. Williams[2], Lucas A. Maddalena[1], Ultan McDermott[2] & Margaret Ashcroft [1✉]

Mitochondria are typically essential for the viability of eukaryotic cells, and utilize oxygen and nutrients (e.g. glucose) to perform key metabolic functions that maintain energetic homeostasis and support proliferation. Here we provide a comprehensive functional annotation of mitochondrial genes that are essential for the viability of a large panel (625) of tumour cell lines. We perform genome-wide CRISPR/Cas9 deletion screening in normoxia-glucose, hypoxia-glucose and normoxia-galactose conditions, and identify both unique and overlapping genes whose loss influences tumour cell viability under these different metabolic conditions. We discover that loss of certain oxidative phosphorylation (OXPHOS) genes (e.g. *SDHC*) improves tumour cell growth in hypoxia-glucose, but reduces growth in normoxia, indicating a metabolic switch in OXPHOS gene function. Moreover, compared to normoxia-glucose, loss of genes involved in energy-consuming processes that are energetically demanding, such as translation and actin polymerization, improve cell viability under both hypoxia-glucose and normoxia-galactose. Collectively, our study defines mitochondrial gene essentiality in tumour cells, highlighting that essentiality is dependent on the metabolic environment, and identifies routes for regulating tumour cell viability in hypoxia.

[1] Department of Medicine, University of Cambridge, Cambridge Biomedical Campus, Cambridge CB2 0QQ, UK. [2] Wellcome Sanger Institute, Hinxton CB10 1SA, UK. [3]Present address: Department of Molecular Life Sciences, University of Zurich, Zurich, Switzerland. [4]These authors contributed equally: Luke W. Thomas, Cinzia Esposito. ✉email: m.ashcroft@medschl.cam.ac.uk

Altered mitochondrial metabolism and function contribute to several pathologies, including cancer. Tumour micro-environmental conditions, such as hypoxia, and changes in nutrient availability can profoundly impact mitochondrial activity, providing metabolic adaptive responses that enable tumour cell survival and promote metastasis[1]. Mitochondrial function is essential for supporting tumour cell proliferation, through the generation of ATP via OXPHOS, and the synthesis of precursors for biomass accumulation, such as amino acids, lipids and nucleotides. Agents that target OXPHOS inhibit cell growth across a broad range of tumour cell types[2,3], while inactivating mutations in mitochondrial genes have also been shown to influence tumour cell proliferation, both positively and negatively, depending on the gene and cell context[4,5].

Mitochondrial ATP synthesis and many other mitochondrial metabolic reactions rely on the intracellular availability of molecular oxygen, and it has been estimated that mitochondria account for >90% of cellular oxygen consumption[6]. Thus, mitochondria are key determinants of cell and tissue oxygenation levels and have been shown to regulate the development of intracellular hypoxia[7,8]. Hypoxia is a characteristic feature of the tumour microenvironment, contributing to disease progression, and is associated with treatment resistance[9] and poor prognosis in patients with solid malignancies[10]. Hypoxia also constitutes mitochondrial stress, as a reduction in oxygen availability limits the activity of the respiratory chain[11]. Metazoan cells have evolved several pathways that sense and respond to changes in oxygen levels, such as the hypoxia-inducible factor (HIF) pathway[1,12], which allows them to adapt in order to maintain cellular homoeostasis and survive[1]. Indeed, a reduction in oxygen availability has wide-ranging effects on multiple cellular functions, and our understanding of the genes and pathways that contribute to hypoxic adaptation continues to grow but is incomplete.

The development of CRISPR/Cas9 gene-editing technology has facilitated the investigation of cellular responses to stimuli at a genome-wide scale[13–15]. Here, we performed genome-wide CRISPR/Cas9 deletion screening under different environmental conditions (normoxia-glucose, hypoxia-glucose, and normoxia-galactose) to interrogate the dependency of tumour cells on nuclear-encoded mitochondrial genes (referred to as mitochondrial genes) and non-mitochondrial genes for their survival when oxygen or glucose is abundant or limited. We provide, what we believe to be, the first comprehensive functional annotation of the nuclear-encoded mitochondrial genome, and classify functional modules of genes that are commonly essential across different tumour cell types. We show that proportionally, twice as many mitochondrial genes are common essential genes when compared to the genome as a whole. Under hypoxia, loss of mitochondrial genes, including OXPHOS genes such as *succinate dehydrogenase subunit C* (*SDHC*), improves the growth of U2OS cells, as well as HeLa and MCF7 cells, and downregulated expression of OXPHOS proteins is an intrinsic response of tumour cells to hypoxia. Conversely, switching the carbon fuel from glucose to galactose to drive mitochondrial respiration significantly increases the essentiality of certain mitochondrial genes for the survival and proliferation of tumour cells. In addition, we show that loss of genes involved in energy-consuming processes that are energetically demanding, such as translation and cytoskeleton arrangement, improve the viability of tumour cells under either hypoxia or galactose. Our study provides a comprehensive survey of mitochondrial gene essentiality in tumour cells, which shows that gene essentiality depends on the metabolic context, and highlights routes for tumour cell viability in hypoxia.

## Results

**The essentiality of mitochondrial genes in tumour cells.** Mitochondria play an essential role in maintaining bioenergetic and biosynthetic homoeostasis to support cell division. While mitochondria contain a well-characterised genome encoding 13 proteins essential for mitochondrial function, there are hundreds of nuclear genes that also encode mitochondrial proteins[16], but their role in tumour cell survival is less well understood. In order to better understand the essentiality of mitochondrial genes for the survival of tumour cells, first, we analysed the gene essentiality scores from the Broad Institute's Achilles Project[17,18], which provides data on genome-wide CRISPR loss-of-function screens, covering 18,333 genes (excluding the 13 protein-coding genes from the mitochondrial genome) in 625 genomically characterised cancer cell lines (as of 19Q3 data release)[19]. In this dataset, common essential genes are defined as a gene that, in a large, pan-cancer screen, ranks in the top X most depleting genes in at least 90% of cell lines. X is chosen empirically using the minimum of the distribution of gene ranks in their 90th percentile least depleting lines. We sorted the essentiality data for mitochondrial genes using the MitoCarta 2.0 database of genes encoding mitochondrial-localised proteins[20], totalling 1137 of the 1158 MitoCarta 2.0 genes (Fig. 1a, Supplementary Data 1), which excludes 22 genes not targeted in the Achilles Project sgRNA library. To gain a comprehensive understanding of the mitochondrial genes and pathways that are most essential for tumour cell survival, we next manually functionally annotated each gene according to published data, which we believe to be the first such full annotation of the mitochondrial proteome (Fig. 1a, Supplementary Data 1). We were able to assign a functional annotation for 91% (1041) of all MitoCarta 2.0 genes (Fig. 1a). Interestingly, a greater proportion of mitochondrial genes (23.2%) were common essential genes, as compared to the proportion of common essential genes in the genome as a whole (11.5%, Fig. 1b, Supplementary Data 1), in line with other studies of this kind[14]. This analysis identified that the largest groups of common essential mitochondrial genes are involved in respiration, mitochondrial gene expression and mitochondrial functions that are well characterised as being required for the survival and proliferation of tumour cells including mitochondrial import and sorting (e.g. *CHCHD4*[3,21–23]) (Fig. 1c). Our functional annotation also allowed us to identify a number of functional modules of genes that represent common essential pathways, such as phosphate metabolism (*PPA2*, *SLC25A3*), and carbonic acid metabolism (*CA5A*) (Fig. 1c). Thus, through our analysis, we have provided a comprehensive overview of the essentiality of mitochondrial genes in tumour cells, cultured under standard normoxic conditions.

**Genome-wide CRISPR deletion screen under different metabolic conditions.** Next, we performed a genome-wide CRISPR/Cas9 deletion screen under three different metabolic conditions in parallel, to identify mitochondrial and non-mitochondrial genes and pathways that determine tumour cell fitness under tumour relevant conditions (Supplementary Fig. 1a). To model conditions of reduced oxygen and glucose availability, commonly found in solid tumours, we compared gene essentiality in (i) glucose media and normoxia (normoxia-glucose), (ii) glucose media and hypoxia (1% $O_2$) (hypoxia-glucose) and (iii) galactose media and normoxia (normoxia-galactose). Notably, two previous independent studies from the same group have described genome-wide CRISPR/Cas9 screens—the first in galactose which assessed dying cells[14], and the second in hypoxia which assessed viable cells[15]. Both studies used a non-adherent immortalised chronic myelogenous leukaemic (CML) cell line, K562[14,15]. Thus here, we perform the first genome-wide CRISPR/Cas9 deletion screen comparing these three

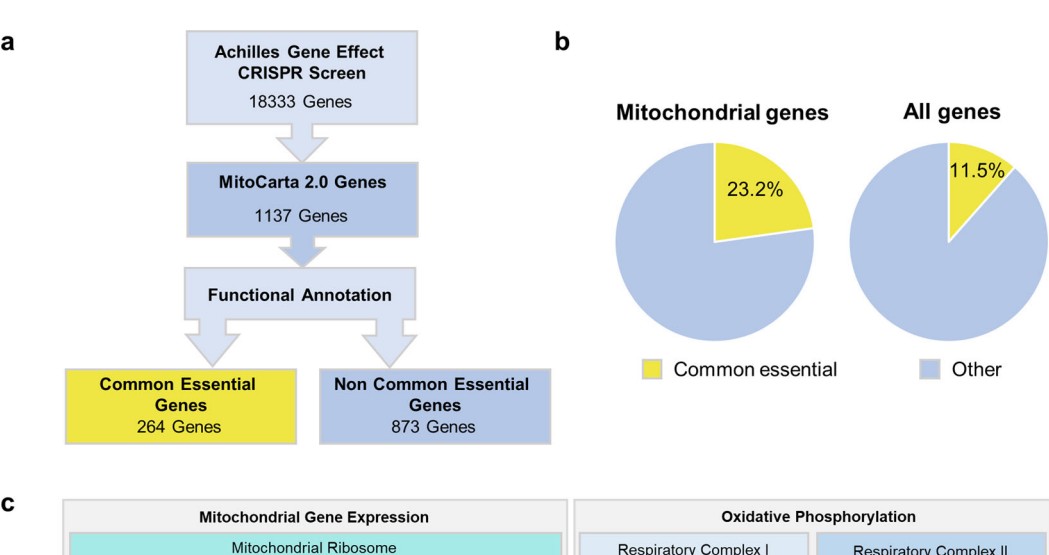

**Fig. 1 The essentiality of mitochondrial genes in tumour cells. a** Schematic diagram shows our analysis pathway for using the Achilles cancer dependency map (DepMap) gene essentiality data. **b** Pie charts show the percentage of common essential genes (shown in yellow) among mitochondrially expressed genes (left chart), and among all genes screened (right chart) in (**a**). Other (non-common essential) genes (shown in blue). **c** List of common essential mitochondrially expressed genes identified in (**a**), grouped by functional annotation. (*) indicates genes with multiple functions.

different metabolic conditions in parallel, using the same screening methodology, and in triplicate. For our study, we selected human osteosarcoma U2OS-HRE-luc cells (referred to as U2OS) originally described by us in a high throughput primary screening context[24]. U2OS cell responses to hypoxia have been well characterised by our group previously[8,21,23,24], and these cells carry no known mutations in the canonical hypoxia-responsive HIF pathway, unlike the CML K562 cell line used in the previous genome-wide CRISPR studies[14,15]. Moreover, the reduction of U2OS cell growth in hypoxia (1% O$_2$) compared to normoxia, is relatively small (Supplementary Fig. 1b), and thus they represent a relevant model for hypoxia-adapted tumours. U2OS cells stably expressing the Cas9 enzyme (U2OS-Cas9) were generated as described (see Materials and Methods), and confirmation of active Cas9 expression was carried out using a dual fluorescent reporter system. Following lentiviral transduction of a genome-wide CRISPR/Cas9 library and puromycin selection (14–15 days), U2OS-Cas9 cells were incubated for 5 days under each condition, harvested and then sequenced for sgRNA abundance (Supplementary Fig. 1a). This time frame of prolonged hypoxia exposure promotes significant changes in hypoxia gene expression (Supplementary Fig. 1c) and results in metabolically adaptive cellular responses. The sequencing reads were analysed using the MAGeCK (version 0.5.6) platform to identify significantly enriched or depleted genes between respective conditions[25] (Supplementary Data 2–4).

In our initial analysis we verified the success of the experimental design by comparing the relative abundance of sgRNAs in U2OS-Cas9 cells transduced with the sgRNA library compared to plasmid only for each replicate under each condition (Supplementary Fig. 2a–f). Analysis of the most significantly depleted genes (false-discovery rate (FDR) <30%) in all three replicates (Supplementary Data 2) identified overrepresentation of genes involved in essential cellular processes such as mRNA processing, cell cycle progression, and OXPHOS (Supplementary Fig. 2g). There was direct corroboration between the over-represented genes in our depleted gene list, and the common essential genes identified by the Achilles project (compare Supplementary Fig. 2g, h), giving us confidence in our screening strategy. Conversely, the most significantly enriched sgRNAs targeted known tumour suppressors and pro-apoptotic genes, such as *NF2*, *TP53* and *NOXA1* (Supplementary Fig. 3).

**Hypoxia promotes the loss of mitochondrial genes and genes involved in energy-consuming processes**. Next, we compared sgRNA abundance in U2OS-Cas9 cells cultured for 5 days in hypoxia-glucose compared to cells cultured in normoxia-glucose (Fig. 2a). First, we verified the success of the sgRNA transduction in hypoxia by comparing the abundance of individual sgRNAs in cells transduced with sgRNA library to those transduced with plasmid only (Supplementary Fig. 2b, e). Of the most significantly depleted genes (FDR < 30%) across all three replicates (Supplementary Data 3), our analysis identified overrepresentation of genes from pathways regulating essential cellular functions e.g. cell cycle, and RNA processing (Supplementary Fig. 4a), similar to cells cultured in normoxia (Supplementary Fig. 2g). Compared to normoxia, we identified 179 genes with significantly enriched sgRNAs in hypoxia, and 18 genes with significantly depleted sgRNAs (FDR < 30%) (Fig. 2b). This latter observation was not surprising, as we have found that U2OS cells tolerated exposure to prolonged hypoxia, as indicated by the small effect of hypoxia on the growth of U2OS cells under these conditions (Supplementary Fig. 1b). Interestingly, 18.4% (33/179) of the sgRNAs significantly enriched in hypoxia targeted mitochondrial genes (Fig. 2c), including several subunits of the respiratory complexes, and genes that regulate the expression of genes from the

mitochondrial genome (Fig. 2d, Supplementary Table 1). However, only 5.5% (1/18) of the sgRNAs significantly depleted in hypoxia targeted mitochondrial genes (Fig. 2c). Moreover, at least 9 of the 33 mitochondrial genes that had significantly enriched sgRNAs in hypoxia were significantly depleted (FDR < 30%) in U2OS-Cas9 cells cultured in normoxia (Fig. 2d, Supplementary Table 1), collectively demonstrating that the essentiality of mitochondrial genes is context-specific, and influenced by environmental oxygen conditions. Notably, culturing U2OS or HCT116 cells in hypoxia for 5 days led to a reduction in the expression of a panel of subunits of respiratory complexes (C) I–IV involved in cell fitness[21] concurrent with an increase in the expression of HIF targets such as BNIP3 (Fig. 2e, Supplementary Fig. 4b). Our data support the idea that suppression of mitochondrial gene expression is an intrinsic response to prolonged exposure to hypoxia, conferring a fitness advantage to cells. The downregulation of mitochondrial function under hypoxia has previously been reported[26]. Also, downregulation of respiration in renal carcinoma cells has been attributed to the action of the HIF pathway on the activity of MYC and PGC1α, which regulate mitochondrial biogenesis[27,28]. Interestingly, we found no concomitant reduction in mtDNA copy number after 5 days of hypoxic culture in U2OS, or HCT116 cells (Supplementary Fig. 5a). Indeed, much research into the cellular response to hypoxia has focused on the HIF pathway and its regulation through hydroxylation by oxygen-dependent dioxygenases. However, none of the HIF pathway genes, including key regulators (e.g. *EGLN1-3* and *VHL*) had significantly altered sgRNA abundances under hypoxia compared with normoxia (Supplementary Fig. 5b), suggesting that the viability of U2OS cells under these culture conditions does not appear to be influenced by loss of components of the HIF pathway. Importantly, these findings corroborate a recent study that also did not identify HIF pathway genes as being essential for the growth of K562 cells[15] which carry a mutation in the *EPAS1* (HIF-2α) gene[18] that is predicted to lead to a truncated protein product lacking the C-terminal oxygen-dependent domain.

Besides identifying genes involved in OXPHOS, overrepresentation analysis of the data from our CRISPR/Cas9 deletion screen in hypoxia-glucose compared to normoxia-glucose also identified significant (FDR < 30%) enrichment of sgRNAs targeting non-mitochondrial genes involved in mRNA processing and regulation of the actin cytoskeleton (Fig. 2f). Further analysis using protein interaction databases identified three major interacting nodes of genes (Supplementary Fig. 5c): a cluster of mitochondrial genes, a cluster of seven genes involved in mRNA processing (Fig. 2g), and a cluster of genes involved in cytoskeleton arrangement, including actin polymerisation (*ARPC2*, *FNBP1*, *CDC42*) and centromere attachment to microtubules (*INCENP*, *KIF18A*) (Fig. 2h). Both transcription/translation and cytoskeleton arrangement are energy-consuming processes that are energetically demanding[29,30], and as hypoxia constitutes an energetic stress, it is likely that loss of these genes improves cellular energy homoeostasis and promotes fitness when oxygen is limiting. Downregulation of transcription and translation is already a well-characterised cellular response to hypoxia, signalled by a reduction in the ATP/ADP ratio[31,32], but precisely how the genes identified here are involved in the cellular response to hypoxia has yet to be described.

**Loss of SDHC improves tumour cell growth in hypoxia**. The most significantly enriched sgRNAs in hypoxia-glucose compared with normoxia-glucose treated cells targeted *SDHC*, a subunit of respiratory complex II, and an enzyme of the tricarboxylic acid (TCA) cycle (Fig. 3a). Therefore, we decided to further validate *SDHC* to demonstrate the influence of mitochondrial gene

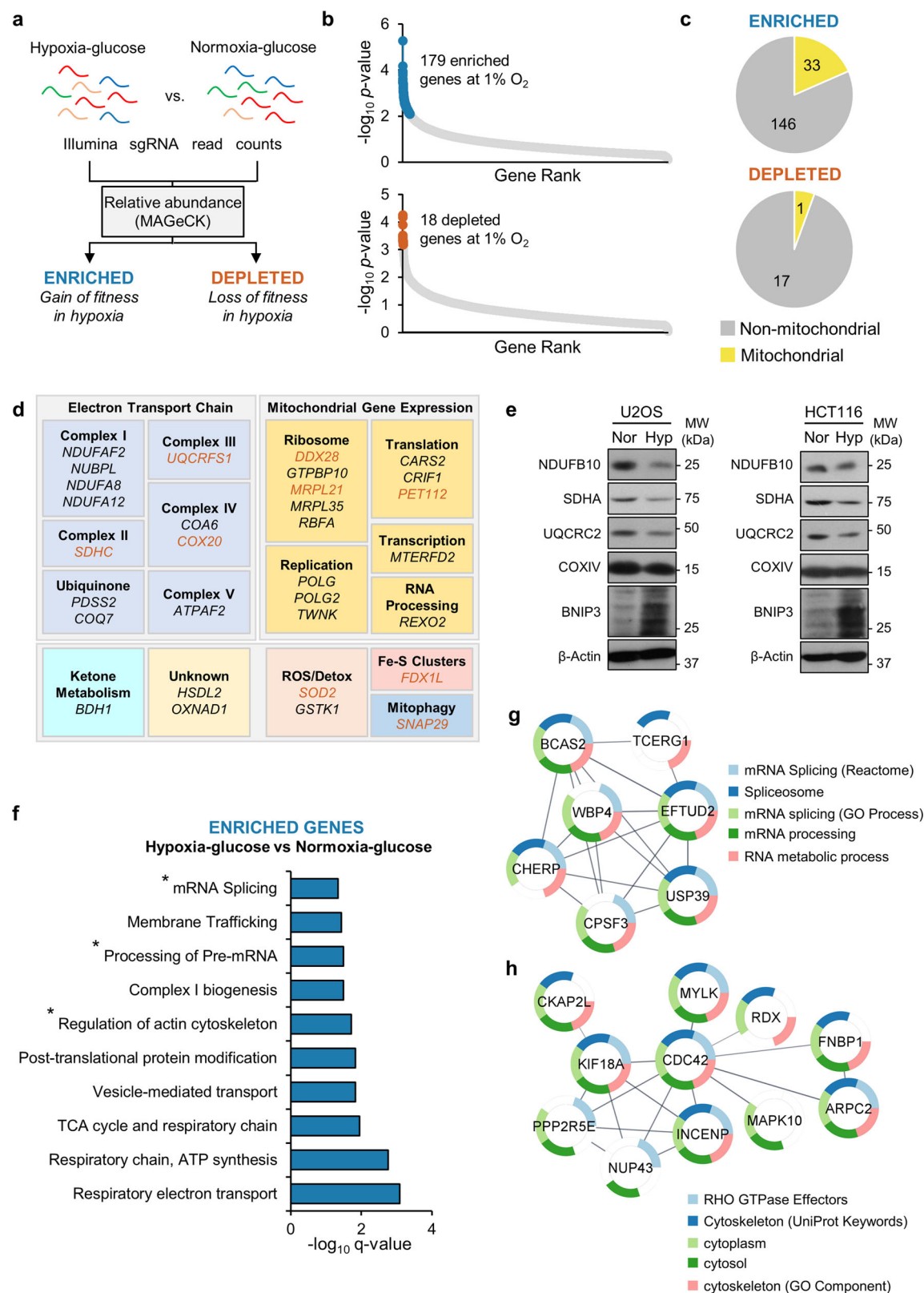

expression on tumour cell growth in hypoxia. Silencing of *SDHC* (Supplementary Fig. 6a) improved the growth of U2OS cells in hypoxia (Fig. 3b), demonstrating that the suppression of mitochondrial function under hypoxia represents a survival response. We confirmed these findings in two other tumour cell lines of different aetiologies, namely MCF7 (breast adenocarcinoma) and HeLa (cervical carcinoma) cells (Supplementary Fig. 6b).

Interestingly, in the colon carcinoma cell line HCT116 where there is a significant reduction in growth in hypoxia, knockdown of *SDHC* did not improve proliferation in hypoxia, and instead significantly reduced cell growth in hypoxia (Supplementary Fig. 6b). In addition, we found that knockdown of *SDHC* in HeLa and HCT116 cells led to a significant reduction in cell growth in normoxia with a trend in reduction in cell growth in MCF7 cells

**Fig. 2 Hypoxia promotes loss of mitochondrial genes and genes involved in energy-consuming processes. a** Schematic diagram shows which datasets were compared in our analysis (hypoxia-glucose vs. normoxia-glucose). Sequencing reads from triplicate incubations were analysed by the MAGeCK analysis platform, and relative sgRNA abundances were calculated between experimental conditions. **b** Charts show FDR-corrected significance values of all sequenced genes, with significantly enriched (blue circles) or depleted (brown circles) sgRNAs in cells cultured in hypoxia-glucose compared to normoxia-glucose. $n = 3$. $p < 0.05$, FDR < 30%. **c** Pie charts show number of mitochondrial genes among the genes identified with significantly enriched or depleted sgRNAs from cells in (**a**). **d** Panel shows 31 selected mitochondrial genes with significantly enriched sgRNAs from (**b**). Genes significantly depleted in normoxia-glucose (plasmid vs. library) are highlighted in brown. **e** Western blots show expression of NDUFB10, SDHA, UQCRC2, COXIV, and BNIP3 in U2OS and HCT116 cells incubated for 5 days in normoxia (Nor) or hypoxia (Hyp, 1% $O_2$). β-Actin used as a loading control. **f** Chart shows overrepresentation analysis of all genes with significantly enriched sgRNAs in hypoxia (hypoxia-glucose vs. normoxia-glucose). Gene sets involving translation (mRNA splicing and processing of pre-mRNA), and regulation of actin cytoskeleton are highlighted (*). **g**, **h** Schematic diagram shows clusters of interacting genes with significantly enriched sgRNAs in hypoxia (hypoxia-glucose vs. normoxia-glucose) involved in mRNA processing (**g**), and cytoskeleton arrangement (**h**).

(Supplementary Fig. 6b). Notably, we did not observe a reduction in U2OS cell growth in normoxia upon *SDHC* knockdown although *SDHC* had been identified as an essential gene from the Achilles project (Fig. 1c), and was depleted in our CRISPR screen (normoxia-glucose, Supplementary Data 2). Collectively these data indicate that U2OS cells tolerate knockdown but not knockout of SDHC in normoxia-glucose. We found that in U2OS cells, SDHC expression was significantly reduced after 5 days of hypoxic culture at both the protein level (Fig. 3c, d) and transcript level (Fig. 3e), in line with the reduced expression of other subunits of respiratory chain complexes (Fig. 2e). Thus, down-regulation of SDHC (and other respiratory complex subunits) appears to be an intrinsic response to hypoxia, which provides a proliferative advantage to cells. Indeed, we found that replacing glucose with galactose—which forces the cells to utilise OXPHOS for the generation of ATP and thus increases basal oxygen consumption rate (OCR) (Supplementary Fig. 6c, d) and reduces lactate production (Supplementary Fig. 6e)—abrogated the protective effect of *SDHC* silencing in hypoxia (Fig. 3f). Furthermore, in the presence of galactose, *SDHC* silencing significantly reduced the growth of cells, when compared to control siRNA treated cells, in both normoxia and hypoxia (Fig. 3f), and interestingly, this reduction in cell growth was more significant in hypoxia compared to normoxia (Fig. 3f). Taken together, our data demonstrate the contextual nature of mitochondrial gene essentiality and suggest that genetic loss of mitochondrial function improves cell fitness in hypoxia when there is glucose to support non-oxidative ATP production.

**Galactose increases mitochondrial gene essentiality.** Since hypoxia suppressed mitochondrial respiratory complex subunit protein expression (Fig. 2e), and led to a reduction in mitochondrial gene essentiality (Fig. 2d), next, we investigated whether stimulating increased mitochondrial activity reversed these changes. Therefore, we repeated the genome-wide CRISPR/Cas9 deletion screen, with U2OS-Cas9 cells cultured in normoxia, in either glucose or (glucose-free) galactose media for 5 days prior to sgRNA sequencing and analysis (Supplementary Fig. 1a, Fig. 4a, Supplementary Data 4). Galactose forces cells to rely on OXPHOS to maintain ATP homoeostasis, since metabolism of galactose to pyruvate leads to no net ATP generation, unlike the metabolism of glucose. When comparing sgRNA abundance in U2OS-Cas9 cells incubated in normoxia-galactose versus normoxia-glucose (Fig. 4a), 32% (59/187) of the most significantly (FDR < 30%) depleted sgRNAs targeted mitochondrial genes (Fig. 4b), including 29 subunits and assembly factors of respiratory chain complexes, as well as genes involved in the TCA cycle, and diverse other metabolic processes (Fig. 4c). Importantly, few (6/188) of these genes were significantly depleted in U2OS-Cas9 cells cultured in normoxia-glucose, when comparing cells transduced with sgRNA library to cells transduced with plasmid control

(Fig. 4c, highlighted in brown). Furthermore, of the significantly enriched sgRNAs in cells cultured in normoxia-galactose compared to normoxia-glucose, 8% (13/169) targeted mitochondrial genes (Fig. 4b). Collectively, this demonstrates that replacing glucose with galactose increases the essentiality of mitochondrial genes from different pathways in tumour cells, highlighting their increased reliance on mitochondrial function for fitness under these conditions. Indeed, U2OS cells were more sensitive to the complex I inhibitor rotenone when cultured in normoxia-galactose compared to normoxia-glucose (Fig. 4d), further supporting the increased essentiality of mitochondrial function in the absence of glucose. Interaction analysis of the genes with the most significantly depleted sgRNAs identified OXPHOS genes as the largest interacting node (Supplementary Fig. 7a). In addition, this analysis identified a related node comprising three genes involved in ROS detoxification (*GPX3*, *SELO*, *TXNRD2*) (Supplementary Fig. 7a), which likely become more essential since mitochondrial respiration and ROS production are concomitantly increased in the presence of galactose.

Interestingly, when comparing significantly enriched sgRNAs in hypoxia-glucose with those significantly enriched sgRNAs identified in normoxia-galactose, we identified several overlapping genes. Indeed, as in hypoxia (Fig. 2f), we found that many of the most significantly enriched sgRNAs in galactose (compared to normoxia-glucose) targeted genes involved in mRNA processing and translation (Fig. 4e). Interaction analysis identified a node of genes including ribosomal subunits (*RPL6*, *RPS18*) and elongation factors (*EEF2*, *EIF3I*) involved in RNA processing (Fig. 4f). Furthermore, there was also significant enrichment of sgRNAs targeting genes involved in cytoskeleton arrangement (Supplementary Fig. 7b), including genes involved in centromere attachment to the cytoskeleton (*INCENP*, *KIF18A*), and actin polymerisation (*ACTR3*, *ARPC4*) (Supplementary Fig. 7b). All four genes were also identified as significantly depleted sgRNAs in normoxia-glucose (Supplementary Data 2), and are classed as common essential genes by the Achilles project[17,18]. As both galactose and hypoxia constitute energetic stress, loss of genes involved in these energy-consuming processes likely helps cells maintain energetic homoeostasis under both conditions. Over-representation analysis of the genes with significantly enriched sgRNAs also identified enrichment of genes involved in glycolysis and gluconeogenesis (Fig. 4e). We identified eight genes are involved in hexose carbon metabolism (Fig. 4g, Supplementary Fig. 7c), including six genes involved in glycolysis (*GPI*, *LDHD*, *PGAM*, *PGK1*, *PKM*, *TPI1*) and two involved in the pentose phosphate shunt (*G6PD*, *PGD*) (Fig. 4g, Supplementary Fig. 7d). Indeed, *PGAM* and *GPI* were the two genes whose sgRNAs were most significantly enriched in normoxia-galactose compared to normoxia-glucose (Fig. 4g). Metabolism of galactose for use in OXPHOS first involves its conversion to glucose-6-phosphate (G6P), followed by metabolism via the canonical glycolysis

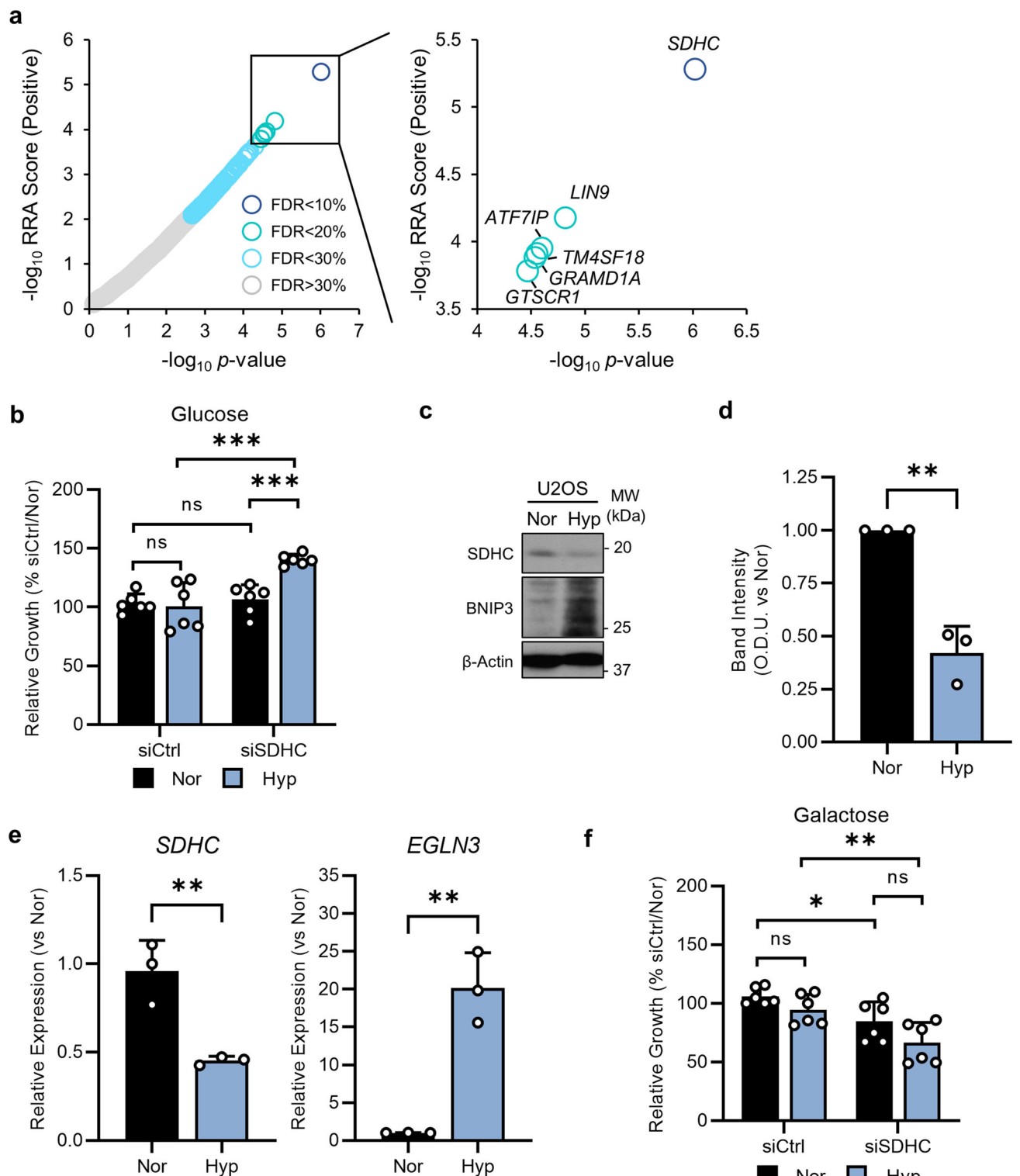

**Fig. 3 Loss of SDHC improves tumour cell growth in hypoxia. a** Chart shows all genes with significantly enriched sgRNAs (robust rank aggregation (RRA) score, positive) in hypoxia-glucose at different FDR thresholds. **b** Chart shows relative growth of U2OS cells cultured for 5 days in normoxia or hypoxia (1% $O_2$) treated with non-targeting control siRNA (siCtrl) or siRNA targeting *SDHC* (siSDHC). $n = 3$. Mean ± S.D.; n.s. not significant; ***$p < 0.001$. **c** Western blots show expression of SDHC, and BNIP3 in U2OS cells incubated for 5d in normoxia (Nor) or hypoxia (Hyp, 1% $O_2$). β-Actin used as a loading control. **d** Chart shows the relative density of SDHC bands from samples in (**c**). $n = 3$; mean ± S.D.; **$p < 0.01$. **e** Charts show expression of *SDHC* and *EGLN3* from U2OS cells incubated for 5d in normoxia or hypoxia (1% $O_2$). $n = 3$; mean ± S.D.; **$p < 0.01$. **f** Charts show relative growth of U2OS cells incubated for 5d in normoxia or hypoxia (1% $O_2$), in the presence of 25 mM galactose, treated with non-targeting control siRNA (siCtrl) or siRNA targeting *SDHC* (siSDHC). $n = 3$; mean ± S.D.; n.s. not significant; *$p < 0.05$; **$p < 0.01$.

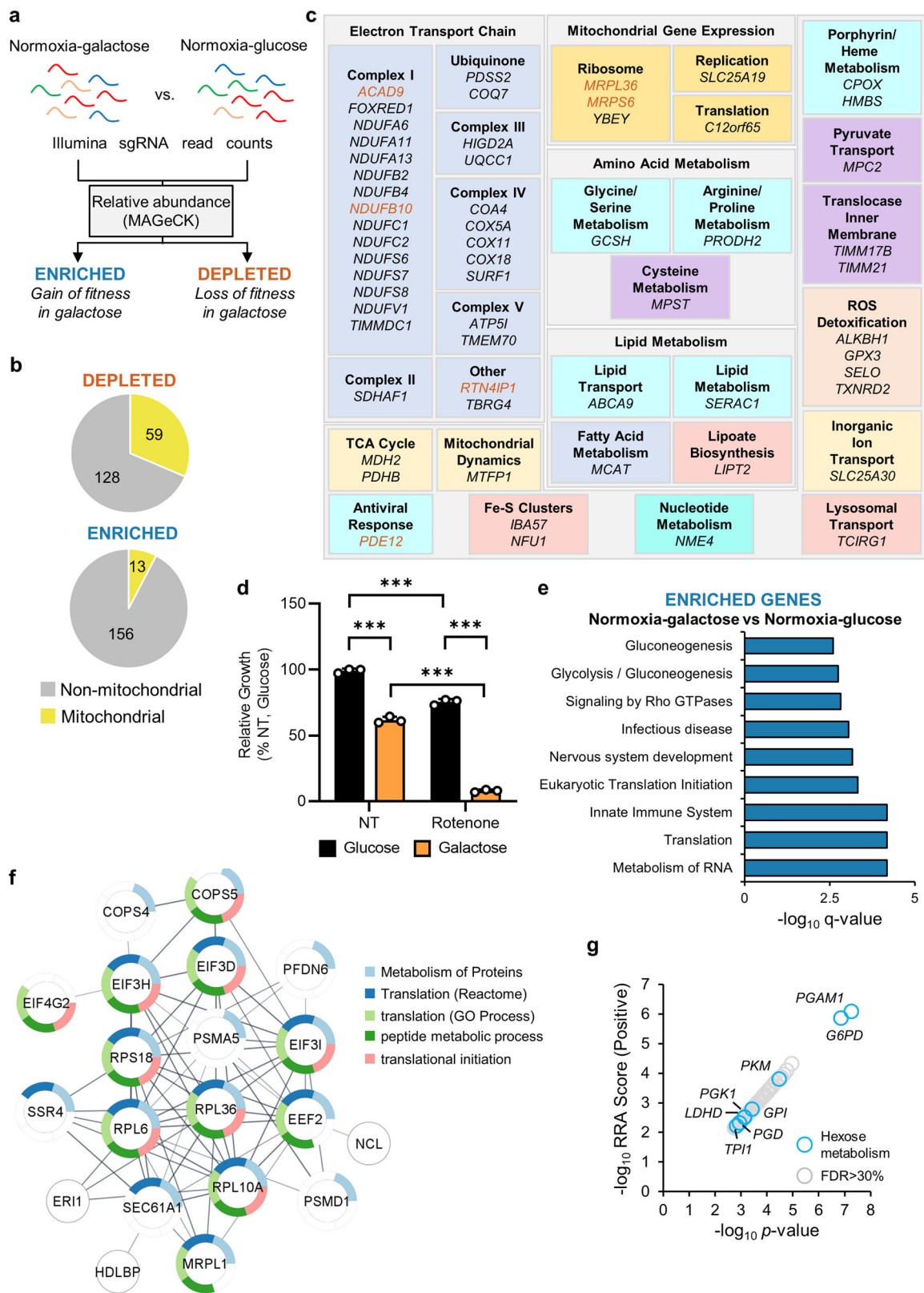

pathway (Supplementary Fig. 7d). A reduction in the diversion of galactose-derived G6P to the pentose phosphate pathway may help maximise ATP production through OXPHOS to support survival/proliferation, as well as loss of *LDHD* to reduce pyruvate fermentation to lactate. However, it is unclear why the loss of genes involved in glycolysis would also be beneficial, though it may be that this conserves the intracellular pool of glucose for

other processes, or promotes the metabolism of alternative carbon sources, such as glutamine, to support OXPHOS.

## Discussion

Mitochondria are involved in regulating eukaryotic cell survival and growth, through their essential roles in controlling

**Fig. 4 Galactose sensitises cells to loss of mitochondrial genes and promotes loss of genes involved in energy-consuming processes. a** Schematic diagram shows which datasets were compared in our analysis (normoxia-galactose vs. normoxia-glucose). Sequencing reads from triplicate incubations were analysed by the MAGeCK analysis platform, and relative sgRNA abundances were calculated between experimental conditions. **b** Pie charts show number of mitochondrial genes among genes with significantly depleted or enriched sgRNAs from cells in normoxia-galactose compared to normoxia-glucose. **c** Panel shows all mitochondrial genes with significantly depleted sgRNAs from (**b**). Genes significantly depleted in normoxia-glucose (plasmid vs. library) are highlighted in brown. **d** Chart shows relative growth of U2OS cells incubated in 25 mM normoxia-glucose or normoxia-galactose, and left untreated (NT) or treated with 31.25 nM rotenone for 72 h. n = 3; mean ± S.D.; ***p < 0.001. **e** Chart shows overrepresentation analysis of all genes with significantly enriched sgRNAs from (**a**). **f** Schematic diagram shows node of selected interacting genes with significantly enriched sgRNAs from (**a**) involved in mRNA processing. **g** Chart shows all genes with significantly enriched gRNAs (robust rank aggregation (RRA) score, positive) from (**a**). Hexose metabolism genes highlighted in blue.

bioenergetic and biosynthetic homoeostasis. Here, we show the essentiality of (nuclear-encoded) mitochondrial genes is context-specific, and that tumour-relevant microenvironmental conditions such as hypoxia or changes in nutrients (e.g. glucose), alters the dependency of tumour cells on mitochondrial genes for their survival and growth. Our findings corroborate those described in two independent studies in which hypoxia[15] and galactose media[14] were used in genome-wide CRISPR/Cas9 screens to interrogate gene essentiality, but which used different methodologies from our study and different cells[14,15]. Several of the mitochondrial genes whose loss improved U2OS cell fitness in hypoxia have been classed as common essential genes by the Achilles Project and were significantly depleted in normoxia in our CRISPR/Cas9 deletion screen. The finding that mitochondrial (and non-mitochondrial) gene essentiality is context-dependent is a conclusion borne out by other gene deletion studies under different conditions, such as metabolic stress or drug treatments[33]. We confirmed the results of our screening data by silencing SDHC, and found that the growth of U2OS, MCF7, and HeLa cells was improved under hypoxia. However, this effect was not observed in HCT116 cells, which have a basal OCR twice that of U2OS cells[3], and whose growth we found was most impaired by hypoxia (Supplementary Fig. 6b). Thus, the improvement in fitness in hypoxia from the loss of SDHC and other OXPHOS genes may be dependent on the degree to which tumour cell proliferation is coupled to mitochondrial function. Several other OXPHOS regulating genes were also significantly enriched in cells from our screen in hypoxia compared to normoxia, including CI subunits, and it was recently shown that single-gene deletions of CI subunits improve the growth of HEK293T cells in hypoxia compared to normoxia[15]. Notably, these findings also align with observations that U2OS cells were consistently less sensitive to low doses of the CI inhibitor rotenone in hypoxia compared to normoxia (Supplementary Fig. 8). Indeed, we confirmed that a reduction in certain OXPHOS proteins occurs in response to chronic hypoxia, a response that has previously been reported to maintain energetic and biosynthetic homoeostasis when mitochondrial activity is impaired[34]. While our study corroborates the recent observation that loss of OXPHOS genes improves cell fitness in hypoxia[15], we found little gene to gene agreement with respect to OXPHOS genes with the previous CRISPR/Cas9 hypoxia study[15]. However, while this may potentially signify some gene specificity between cell lines used in different CRISPR screens, it is important to consider that in general, gene by gene differences between screens is a characteristic of large scale screens of this kind due to the variability of gene rank and significance between individual experiments. Thus here, we performed three independent biological repeat screens of all metabolic conditions (Supplementary Fig. 1a, Supplementary Data 2–4), and our subsequent analysis was performed only on those significantly depleted or enriched genes for each condition that we identified on all three independent biological repeat screens. Together with the use of a second generation vector

backbone as outlined here, and importantly for statistical considerations, having at least five gRNA per gene in the CRISPR library[35] increases the statistical power of each experiment.

Interestingly, compared to normoxia, we identified that loss of several genes involved in mRNA processing and translation, as well as cytoskeleton regulation, improved the fitness of cells in both hypoxia and galactose. Protein translation is an energetically demanding process, and the downregulation of global translation in hypoxia has been well characterised and linked to changes in intracellular ADP/ATP ratios[31,32]. Cytoskeleton rearrangement is a similarly energetically demanding process: for example, actin polymerisation requires the hydrolysis of a molecule of ATP for every addition of an actin monomer[30]. Thus, the improvement in tumour cell viability through the loss of actin polymerisation genes, and other cytoskeleton arrangement genes, may also be due to the benefit this provides to bioenergetic homoeostasis. It will be interesting to investigate whether changes in ATP availability in hypoxia (or in the presence of galactose) stimulate signalling pathways that regulate cytoskeleton arrangement processes in a similar manner to protein translation.

Our study did not find that peroxisome-related genes were essential for the survival of U2OS cells in hypoxia, which was described in a genome-wide CRISPR/Cas9 hypoxia screen using K562 cells[15]. However, this peroxisome observation from the CRISPR screen could only be verified in one cell line of four tested, and could not be verified or recapitulated in the K562 cells used for the screen[15]. This highlights that reproducibility and cell to cell variability is an important consideration when interpreting the results of genome-wide CRISPR screens of this type. Importantly, here, we sought to address this by comparing our results with the pan-cancer gene essentiality data provided by the Broad Institute's Achilles project. Although this comparison has limited applicability due to differences in methodology, conditions and analysis pipelines, we were able to show from our overrepresentation analyses that there was direct corroboration in our depleted gene list, and the common essential genes identified by the Achilles Project (compare Supplementary Fig. 2g, h). Nevertheless, we cannot exclude the possibility that some of the results presented here are specific for U2OS cells. Indeed, while we found that cytoskeletal gene depletion was beneficial for U2OS cell growth in both hypoxia and galactose, this was not the case in K562 cells used in previous galactose and hypoxia screens described by the Mootha group[14,15]. This may represent a difference in cytoskeleton gene essentiality between adherent (U2OS) and non-adherent (K562) cells.

Together, our study, to the best of our knowledge, provides novel observations regarding the contextual nature of mitochondrial and non-mitochondrial gene essentiality in tumour cells and also corroborates previous observations in other cell types. In addition, different genes and pathways have been identified that warrant further investigation for their utility as therapeutic targets for both normoxic and hypoxic tumour cells. Our study also cautions that the development of therapeutic

agents that target certain cellular processes, such as mitochondrial metabolism or translation, must take into consideration that their efficacy will depend on microenvironmentally induced adaptations in tumour cells, such as those mediated by hypoxia.

## Materials and methods

**Cell culture.** Human U2OS osteosarcoma, MCF7 breast carcinoma, HeLa cervical cancer, and HCT116 colon carcinoma cell lines were all obtained from American Tissue Culture Collection (ATCC). All cell lines were maintained in Dulbecco's modified eagle medium ('standard' DMEM, #41965062, ThermoFisher Scientific) containing glucose (4.5 g/L), and supplemented with 10% foetal calf serum (FCS, SeraLabs), penicillin (100 IU/mL), streptomycin (100 µg/mL) and glutamine (6 mM), all purchased from ThermoFisher Scientific. Cell lines used were authenticated, and routinely confirmed to be negative for any mycoplasma contamination. Hypoxia was achieved by incubating cells in 1% $O_2$, 5% $CO_2$ and 94% $N_2$ in a Don Whitley H35 workstation, without agitation.

**Generation of U2OS-Cas9 expressing cells.** Human U2OS osteosarcoma cells (U2OS-HRE-Luc cells[24]) were transduced with lentivirus prepared from the pKLV2-EF1a-Cas9Bsd-W plasmid in the presence of polybrene (8 µg/mL)[19]. Following transduction, U2OS-Cas9 expressing cells were selected with blasticidin (20 µg/mL), and Cas9 expression and activity were confirmed. Briefly, to assess the ability of the U2OS-Cas9 expressing cells to efficiently silence full-length gene expression, cells were transduced with a lentivirus produced from the Cas9 reporter vector, pKLV2-U6gRNA5(gGFP)-PGKBFP2AGFP-W which was a gift from Kosuke Yusa (Addgene plasmid#67980; http://n2t.net/addgene:67980; RRID: Addgene_67980)[36]. This vector contains BFP and GFP expressing cassettes, as well as an sgRNA targeting GFP—efficient Cas9 activity would therefore be expected to result in silencing of GFP signal. The ratio of BFP-only and GFP-BFP-double-positive cells was analysed on a BD LSRFortessa instrument (BD Biosciences), 3 days post lentiviral transduction. The data were subsequently analysed using FlowJo (BD Biosciences). Efficiently transduced U2OS-Cas9 cells showed high BFP expression but the loss of GFP signal, indicating that the majority of the U2OS-Cas9 cells expressed active Cas9.

### Genome-wide CRISPR/Cas9 screen

*Viral infection.* The U2OS-Cas9 cells were lentivirally transduced with a genome-wide CRISPR library of 90,709 sgRNAs targeting 18,009 human genes, at coverage of at least five sgRNAs per gene[36]. In total, $3 \times 10^7$ cells were infected with lentiviral-containing supernatant at an MOI of 0.3, allowing one sgRNA integration per cell and therefore aiming to provide coverage of >100 cells expressing each sgRNA. The lentiviral transduction of the U2OS-Cas9 cells with the CRISPR library was performed independently three times, for three independent repeat screens. Each independent repeat screen assessed three conditions in parallel, normoxia-glucose, hypoxia-glucose and normoxia-galactose (Supplementary Fig 1a).

*Puromycin selection and passaging.* Two days after CRISPR library transduction, cells were exposed to puromycin (2 µg/mL) for at least 10 days, then the U2OS-Cas9 library cells were analysed on a BD LSRFortessa instrument (BD Biosciences) to confirm >90% BFP positivity. The U2OS-Cas9 library cells were then selected with puromycin for a further 4–5 days during expansion, and at least $20 \times 10^6$ cells were carried forward after each passage to maintain the integrity of the CRISPR library. Cells were cultured in Nunc® T175 triple flasks, and cell passaging was carried out by washing with 22 mL of phosphate-buffered saline (PBS, ThermoFisher #10010056), dissociated with 12 mL of 0.5% trypsin–EDTA (ThermoFisher #25300096), and resuspended in 18 mL of media (DMEM, ThermoFisher #41965062), per flask.

*Setup of experimental conditions.* After two weeks of puromycin selection and expansion, the U2OS-Cas9 library cells were pooled, counted and at least $210 \times 10^6$ cells were plated into triple flasks ($15 \times 10^6$ cells per triple flask) in 'standard' DMEM, then incubated at 37 °C/5% $CO_2$/normoxia for 24 h. After 24 h, the 'standard' glucose-containing media was replaced with glucose-free galactose-containing media (DMEM, #A14430, ThermoFisher Scientific, supplemented with 4.5 g/L D-(+)-galactose, 4 mM L-glutamine, 1 mM sodium pyruvate, and 10% FCS), in a subset of flasks, to pre-condition the cells. To begin the screen, for each experimental condition (described below), $21 \times 10^6$ U2OS-Cas9 library cells were transferred to 7 separate triple flasks ($3 \times 10^6$ cells per triple flask) and left to adhere overnight under normal culturing conditions (37 °C/5% $CO_2$/normoxia). The following day, the medium was replaced with fresh medium for each condition, and cells were placed in the desired atmospheric conditions (normoxia or hypoxia (1% $O_2$)). Cells were grown for 5 days under the following experimental conditions: (1) cells grown in 'standard' glucose-containing medium in normoxia (normoxia-glucose), (2) cells grown in 'standard' glucose-containing medium in hypoxia at 1% $O_2$ (hypoxia-glucose), and (3) cells grown in glucose-free galactose-containing medium in normoxia (normoxia-galactose). No passaging of cells was carried out for the duration of the screen. Three independent repeat screens were performed, each of which assessed three conditions in parallel, normoxia-glucose, hypoxia-glucose, and normoxia-galactose (Supplementary Fig 1a).

*Cell harvesting.* After 5 days of culture under each experimental condition, U2OS-Cas9 library cells from each condition were independently pooled, counted, pelleted, resuspended in 200 µL of PBS, and stored at −20 °C until further analysis. Cell counts confirmed that at least $20 \times 10^6$ cells harvested from each condition were taken forward for sequencing (Supplementary Fig. 1b).

**Illumina sequencing of sgRNAs and statistical analysis.** Genomic DNA extraction and Illumina sequencing of sgRNAs were conducted as follows[37]. In brief, 72 µg of total extracted DNA was used to set up 36 polymerase chain reactions (PCR, 2 µg each) using 10 µM concentrations of forward and reverse primers (below) following which PCR products were purified using spin columns before a second PCR reaction was carried out to incorporate indexing primers for each sample. DNA was purified using SPRI beads and submitted for Illumina sequencing.

F primer: ACACTCTTTCCCTACACGACGCTCTTCCGATCTCTTGTGGAA AGGACGAAACA

R primer: TCGGCATTCCTGCTGAACCGCTCTTCCGATCTCTAAAGCGCA TGCTCCAGA

Enrichment and depletion of guides and genes were analysed using Robust Ranked Aggregation in the MAGeCK statistical package (ver 0.5.6) by comparing read counts from the Treatment samples with those from the Control samples and using the '--num-goodsgrna min_number' function to filter genes that have less than 2 'good sgRNAs', or sgRNAs that fall below the -p threshold (https://sourceforge.net/projects/mageck/)[25]. As an initial quality control assessment, we compared the Control samples to the Plasmid library for essential depleted genes. We confirmed that the majority of depleted genes were those expressed at higher levels, with almost no depletion for non-expressed genes. Additionally, gene set enrichment analysis of genes based on their depletion scores confirmed that essential biological processes for cell survival were the most significantly depleted in each condition. These included genes from spliceosome, ribosome, DNA replication and RNA polymerase families.

**Antibodies.** Antibodies used were as follows, with dilutions in parentheses: rabbit polyclonal NDUFB10 (ab196019, 1:2000) from Abcam; rabbit polyclonal SDHA (11998, 1:1000) from Cell Signaling Technology; rabbit polyclonal UQCRC2 (ab14745, 1:1000) from Abcam; rabbit polyclonal COXIV (4850, 1:10,000) from Cell Signaling Technology; rabbit polyclonal BNIP3 (HPA003015, 1:2000) from Cambridge Biosciences; rabbit polyclonal SDHC (PA5-79966, 1:1000) from Invitrogen; mouse monoclonal β-Actin (ab6276, 1:10,000) antibody from Abcam; donkey anti-rabbit (NA934, 1:1000) and anti-mouse (NA931, 1:1000) horseradish peroxidase (HRP)-linked secondary antibodies from VWR. Western blot signal intensity was measured per lane using ImageJ (NIH) analysis software. Sample protein band intensities were normalised to the load control protein β-Actin. Relative band intensities were calculated relative to the internal control sample.

**Respirometry.** OCR was determined using a Seahorse XF96 Analyser (Seahorse Bioscience). Respiratory profiles were generated by serial treatment with optimised concentrations of oligomycin (1 µg/mL), p-[trifluoromethoxy]-phenyl-hydrazone (FCCP, 500 nM), and rotenone (500 nM). Cell number normalisation was carried out post-respirometry using sulforhodamine B (SRB) staining of TCA fixed cells in the assay plate.

**Quantitative PCR (QPCR) analysis.** For gene expression analysis and mitochondrial copy number methodology using QPCR, see Supplementary Methods and Supplementary Table 2.

**siRNA transfection, compound treatment and SRB cell growth assays.** Cells were plated in appropriate tissue culture vessels and allowed to adhere overnight prior to siRNA transfection or compound treatment.

*siRNA transfection.* Transfection complexes were made by combining 50 nM siRNA (for siRNA sequences see Supplementary Table 3) with HiPerfect (QIAGEN) transfection reagent in 100 µL serum-free DMEM. Complexes were then diluted in sufficient DMEM containing either 25 mM glucose or galactose (both with 10% foetal calf serum (FCS, SeraLabs), penicillin (100 IU/mL), streptomycin (100 µg/mL) and glutamine (6 mM)) for all samples. Media in wells removed then replaced with transfection complex-containing media for 72 h. The transfection procedure was then repeated, and cells were incubated for a further 48 h (5 days total).

*Compound treatment.* Standard growth media was replaced with compound free DMEM containing either 25 mM glucose or galactose, or containing 31.25 nM rotenone. Cells were then incubated in normoxia or hypoxia (1% $O_2$) for 72 h.

*SRB assay.* At the end of incubation, media was removed, and cells were fixed with 10% (w/v) trichloroacetic acid (TCA) for 30 min. TCA was washed with water, wells were allowed to air dry, and then an excess of 0.4% (w/v) SRB in 1% acetic

acid was used to stain fixed cells for >10 min. Excess SRB was washed off with 1% (v/v) acetic acid solution. Bound SRB was resuspended in a suitable volume of 10 mM Tris, and absorbance of the solution measured at 570 nm.

**Overrepresentation analysis.** Common essential gene lists were accessed from the Broad Institute Cancer Dependency Map data portal (https://depmap.org/portal/download/)[17,18]. 19Q3 data release used throughout, covering 625 cell lines. Overrepresentation analyses were carried out at the Broad Institute analysis portal (http://software.broadinstitute.org/gsea/msigdb/annotate.jsp). FDR q-value set at 0.05.

**Protein–protein association network analysis.** Protein–protein association networks were generated using the stringApp v 1.5.1 in Cytoscape v 3.8.0[38]. Proteins in the networks were grouped based on their STRING interactions using Markov clustering in the clusterMaker2 app (v 1.3.1)[39], with additional manual curation. Clusters were functionally characterised using the stringApp to perform functional enrichment analysis, with the corresponding non-clustered network set as the background.

**Statistical analysis and reproducibility.** A number of replicates ($n$), error bars and $p$ values are described in each figure legend. All statistical analysis of data carried out in Microsoft Excel, apart from analysis of sgRNA sequencing data which was carried out in MAGeCK (version 0.5.5). For growth assays and QPCR, conditions were compared by two-tailed unpaired $t$ tests. For comparison of sgRNA abundances, all genes were ranked for enrichment and depletion separately by the α-RRA method in MAGeCK[25]. Multiple comparison correction was applied using the Benjamini–Hochberg method, following exclusion of genes with fewer than 2 'good' targeting sgRNAs ($p < 0.1$), and an FDR threshold of 0.3 (30%) was applied.

**Reporting summary.** Further information on research design is available in the Nature Research Reporting Summary linked to this article.

## Data availability
Genes identified from the functional annotation of all MitoCarta 2.0 genes represented in Achilles Cancer Dependency essentiality dataset described in Fig. 1, are included in Supplementary Data 1. For our CRISPR/Cas9 screens, the raw read counts, the ranked gene lists (enriched and depleted genes), and MAGeCK analysis data (FDR-corrected $p$ value) for each condition and repeat screen (Supplementary Fig. 1a) are included in the Supplementary Data 2–4. The FDR-corrected significance values for all genes with significantly depleted sgRNAs at different FDR thresholds (<10%, <20%, <30%, >30%) for each condition and repeat screen are included in Supplementary Fig. 2a–c. Raw data for charts (Fig. 3b, d–f, Fig. 4d and Supplementary Fig. 4b) are included in Supplementary Data 5. Raw western analysis data for Fig. 2e and Fig. 3c are included in Supplementary Figs. 9 and 10. For any further information, or reasonable requests, please contact the corresponding author.

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

## Acknowledgements

L.W.T., C.E., R.E.M. and L.A.M. were supported by Addenbrooke's Charitable Trust (900187), Wellcome Trust (RG93172) and Cancer Research UK (C7358/A19442 and RG91141) grants awarded to M.A, who also received funding from the Medical Research Council. U.M. received funding from the European Research Council under the European Union's Seventh Framework Programme (FP7/2007–2013)/ERC synergy grant agreement n° 319661 COMBATCANCER, and together with S.P., J.Y. and S.P.W. was funded by the Wellcome Sanger Institute. Thanks to Francesca M. Buffa and James Hok-Fung Chan (University of Oxford, UK) for helpful discussions on bioinformatics analyses.

## Author contributions

L.W.T. designed and performed experiments, analysed data and co-wrote the paper. C.E. designed and performed the CRISPR/Cas9 screen, performed experiments, analysed data and co-wrote the methodology. R.E.M. assisted with the CRISPR/Cas9 screening effort and provided technical support, and S.P. and J.Y. assisted with library transductions. U.M. assisted the sequencing effort, and together with S.P.W. analysed raw sequencing data. L.A.M. assisted with interactome analyses. M.A. provided the concept for the study, designed experiments, analysed data, co-wrote the paper and acquired funding. C.E., R.E.M., S.P. and U.M. edited the paper. All authors approved the final paper.

## Competing interests

The authors declare no competing interests.
