## [Peer Review File · Communications Biology]

Reviewers' comments:

Reviewer #1 (Remarks to the Author):

This is an interesting screen paper.

It is important that this work be published as it validates and questions previous findings.

I would like the authors to comment how their current findings are similar and different than previous hypoxia and galactose screens.

For example, VHL loss was not a top hit in hypoxia but in the Jain et al Science 2016 paper the whole premise is due to VHL loss..

Reviewer #2 (Remarks to the Author):

In this study, the essentiality of all genes for the cellular response to hypoxia was surveyed using a CRISPR screen, with an enrichment of nuclear encoded mitochondrial genes reported as essential. While the results are interesting, novelty is a problem. There are also some potential statistical issues, and places where clarity could be enhanced.

Major points:

(1) The study appears to be very similar to the recently published paper from Isha Jain & Vamsi Mootha (ref #12). It is therefore incumbent on the authors to delineate what is new and/or different here. It is not enough to simply refer to this paper deep in the results with a throwaway "consistent with a recent study".

Since the results of the other paper are publicly available in the supplemental files, it would be useful to do a direct comparison across all the genes reported here and in the other paper, and provide some figures on the degree of overlap (e.g. Venn diagrams). This would enhance the data set in the current paper. The discussion could then be focused on what came out differently in the two studies.

(2) The essentiality data set covered 1137 of the 1146 genes in MitoCarta. What were the missing ones? One presumes these missing 9 genes were ones that could not be CRISPR'd in the original screen? This needs to be stated.

(3) The paper in a number of places uses qualitative terms or incorrect language which does not enhance the narrative. e.g., line 67 "a large number of" (define large). Line 72 "we have provided" - since the screen was not actually conducted by you this seems to be a bit of a stretch.

(4) It is stated that the experiments were performed in triplicate, but because these are cell culture experiments, it is not clear if these are simply 3 technical replicates (i.e. 3 flasks of cells from the same batch, or maybe different passages of the same cells), or they are true biological replicates (e.g., 3 completely independent stocks of cells prepared at different times). Either way, an N of 3 is at the very lowest end of the scale for statistical rigor. A power analysis should be included, to determine what effect size the study was capable of detecting with this number of replicates.

(5) Line 98. "Good agreement" with the Achilles Project data is not really supported by Suppl. Fig. 1d. The Venn diagram shows that of 264 mito genes in the Achilles set, only 109 (41%) were also found here. Likewise for all genes, only 49% were overlapped. The Venn diagrams should at least be drawn to scale, so that the degree of overlap is accurately represented.

Minor points:

(1) The rationale for selection of U2OS cells needs to be stated and justified.

(2) The seahorse OCR data in Suppl. Fig. 4 are not normalized to anything. The method states the data were normalized to cell count, but the y-axis label does not include this information (i.e. pmols/min/n-cells).

Reviewer #3 (Remarks to the Author):

In the manuscript by Thomas et al, the authors defined mitochondrially essential genes encoded by nuclear genes and performed two genome wide CRISPR/Cas genetic screens under conditions of metabolic stress. The first screen was performed in U2OS cells under hypoxia and the second performed with galactose to promote the use of OXPHOS for ATP production. The hypoxia screen identified SDHC and the galactose screen ROS genes.

While the quality of the writing in the manuscript is good, the Introduction and Discussion are extremely underdeveloped and it lacks an abstract to summarize the study. The formatting of this manuscript is far off what would be expected at Communications Biology. Once the authors address some of the issues mentioned above, I would be happy to review the manuscript.

In the meantime:

1. SDHC is a critical gene for both the TCA cycle and OXPHOS, could the authors elaborate on why this was a top hit in the hypoxia screen?
2. Is SDHC essential across multiple cancer types or where hypoxia plays a key role? Or is its role restricted within sarcoma?
3. Could the authors comment on the inter-relatedness of the two screens?

Response to reviewers' comments, manuscript COMMSBIO-20-1799-T. Thomas et al.

Reviewer #1 (Remarks to the Author):

This is an interesting screen paper.

It is important that this work be published as it validates and questions previous findings.

Thank you for your encouraging comments.

I would like the authors to comment how their current findings are similar and different than previous hypoxia and galactose screens.

Thank you, we have included a new section in results describing our screen in more detail (ln 118-159), and we have commented on the differences between our screen and the two separate previous hypoxia and galactose CRISPR screens (ln 123-130).

The major difference from the previous screens, is that our screen was performed in normoxia, hypoxia and galactose conditions in one study, and the screen was performed 3 independent times. The cells chosen for our screen (U2OS-HRE-luc) were generated from parental U2OS osteosarcoma cells, and originally described by us in a high throughput primary screening context before (Chau et al 2005 Can Res 65(11):4918-28). These cells contain a hypoxia/HRE luciferase reporter, and were used to generate the Cas9 expressing cells for our screen. Importantly, these cells have both a normal HIF response in hypoxia (new Supplementary Fig. 2c), and show enhanced OXPHOS when cultured in galactose (Supplementary Fig. 6c,d). Our screen analysed viable, adherent cells surviving after 5 days in culture conditions, which were harvested and counted prior to PCR/sequencing. These culture conditions are analogous to our small molecule inhibitor screen (Chau et al 2005 Can Res 65(11):4918-28), and subsequent screens which have successfully identified novel (mitochondrial and non-mitochondrial) inhibitors of hypoxia signalling and tumour cell growth.

Arroyo et al 2016. Cell Metab. 24(6): 875–885, only describes a galactose screen; and the very recent study by Jain et al 2020 Cell. 30;181(3):716-727, only describes a hypoxia screen. In these studies, the CRISPR library, screening conditions and cells used were different from ours. These 2 previous screens used an Avana lentiviral genome-wide CRISPR/Cas9 library with only 74,687 single-guide RNAs (sgRNAs) (with up to 4 sgRNAs per gene) targeting 18,335 distinct human genes. Our CRISPR library from the Sanger Institute contains 90,709 gRNAs -5 independent sgRNAs per gene, targeting 18,009 genes, thus more superior coverage per gene.

Furthermore, these 2 previous screens (Arroyo et al 2016, and Jain et al 2020) used a *non-adherent* immortalized chronic myelogenous leukemic cell line -K562. This K562 cell line was also used in a complex III respiratory inhibitor CRISPR screen from the Mootha group described in Jain et al 2016 Science 352(6281): 54–61 (see below). It is not clear from the authors why the K562 cell line was chosen for use in these 3 screens (Arroyo et al 2016. Cell Metab; Jain et al 2016 Science; Jain et al 2020 Cell). But it is particularly note-worthy to point out that the K562 cell line used these 3 studies from the Mootha group (Arroyo et al 2016. Cell Metab; Jain et al 2016 Science; Jain et al 2020 Cell) has been shown to carry mutations in several genes that might perhaps unknowingly affect metabolism and/or the HIF pathways (see https://www.cbioportal.org/patient?sampleId=K562_HAEMATOPOIETIC_AND_LYMPHOID_TISSUE&studyId=ccl_broad_2019). For example, K562 cells carry a mutation in EPAS1/HIF-2A, which is predicted to generate a shortened protein lacking the oxygen dependent domain which would lead to constitutively stabilised HIF-2 α in normoxia, and could have impacted the results of these 3

previous screens. As outlined above, we chose to use U2OS cells for our study for the reasons outlined above and because they carry no known mutations in the canonical HIF pathway genes.

Another difference between our screen and the Arroyo et al 2016 galactose screen, is that the cells were cultured for only 24-hr in glucose or galactose medium, then the *dead* (annexin V +) cells were harvested and sequenced. It is worth noting that moving cells into galactose from glucose causes a significant reduction in cell growth within the first 24 hours irrespective of any modulation of the genome. To avoid dramatic differences between glucose and galactose conditions in our screen, we equilibrated cells in galactose alongside glucose for 24 hours, then replated the cells in equal numbers before performing the screen over 5 days. Thus, our plating conditions and cell density allowed for any potential effects of treatment-induced growth inhibition deleteriously impacting the integrity of the library over the entire duration of the screen. Moreover, we harvested and sequenced the *viable* cells. This is because the premise for our screen was to identify genes involved in promoting survival under different metabolic conditions. Furthermore, Arroyo et al 2016 only performed 2 biological replicates. We performed 3 independent biological repeats in our screen (new Supp Fig 1a), involving 3 independent library infections and subsequent treatments.

Regarding the recently described hypoxia screen by Jain et al 2020 in Cell. This study provides 'essential' gene data only on 21% versus 1% O₂, just as in our study. Consistent with Jain et al 2020, and not a surprising result, we also identified that HIF pathway genes were not depleted (essential) in hypoxia compared to normoxia (Supplementary Fig. 5b). However, we do not identify peroxisomal/lipid metabolism as a top pathway depleted (essential) in our screen (hypoxia compared to normoxia), unlike the Jain 2020 study (Fig 4). It is very important to point out that in the Jain et al 2020 study, the authors could not recapitulate the peroxisome/lipid finding from their screen when they validated these findings in the screening cell line (K562), nor could they recapitulate the peroxisome/lipid finding in other cells lines (e.g. HeLa, MCH58). In fact, they were only able to recapitulate the peroxisome/lipid finding in one cell line (HEK293Ts). In contrast to the Jain et al 2020 hypoxia study, in our study, all the top genes described from our screen were shown to recapitulate in the same cell line as the screen (U2OS), along with two other cell lines (HeLa, MCF7). Thus, completely validating the results from our screen. Another unexpected finding in the Jain et al 2020 hypoxia study, unlike our study, was that there was no change in the essentiality of translation-related genes, and yet it is well known that translation is suppressed in hypoxia in many cell lines.

For example, VHL loss was not a top hit in hypoxia but in the Jain et al Science 2016 paper the whole premise is due to VHL loss..

Thank you. The Jain et al Science 352(6281): 54–61 paper describes a pharmacological complex III inhibitor (antimycin A) screen in normoxia – which is clearly different from our screening conditions. The authors suggest that by inhibiting the respiratory chain in this way, they were ‘mimicking’ mitochondrial disease. While antimycin A will block respiration and (at appropriate dose) inhibit HIF-1 α induction in hypoxia, the Jain et al 2016 study describe neither a hypoxia nor galactose screen. So, from this perspective, it is not experimentally comparable to our study, nor the two studies outlined above. It is also important to point out that out-with its effects on blocking respiration, antimycin A is an agent that causes significant mitochondrial/cell toxicity. It is unclear from the study what is the rationale of how the toxic effects of antimycin A specifically relate to mitochondrial disease. As with the two other Mootha group studies mentioned above, the Jain et al 2016 Science screen used K562 cells.

Given that hypoxia blocks mitochondrial activity, in part, though activation of HIF, and galactose drives mitochondrial activity, neither of these conditions (which are in our screen) are comparable to the antimycin A treatment of the Jain et al 2016 screen. The authors identify von Hippel-Lindau (VHL) factor as the most effective genetic suppressor of death induced by antimycin A treatment (Fig 1D of Jain et al 2016, - they identify VHL as a top hit from their screen when they compare antimycin A treated with no antimycin treatment in normoxia). Also, the Jain et al 2016 study describes that hypoxia protect cells from the deleterious effects of antimycin A.

It is well-established that loss of VHL which leads to constitutive activation of HIF also causes blockade of mitochondrial function in normoxia, and that loss of VHL (e.g. in RCC or mice), or hypoxia leads to reduced sensitivity to respiratory chain inhibition.

- a) Loss of VHL in RCC, causes constitutive HIF activation, and resultant suppression of mitochondrial OXPHOS (Simmonet 2002, Carcinogenesis vol.23 no.5 pp.759–768, 2002; Briston et al 2018, doi: 10.3389/fonc.2018.00388), and reduced CIII activity, which can be rescued by reconstitution of VHL (Hervouet et al 2005 doi:10.1093/carcin/bgi001, also see Zhang et al 2007 DOI 10.1016/j.ccr.2007.04.001) or knockdown of HIF-2 α (Zhang et al 2007 DOI 10.1016/j.ccr.2007.04.001; Briston et al 2018 doi: 10.3389/fonc.2018.00388)
- b) Loss of VHL (-/-) rescues deleterious effects of loss of the mitochondrial regulator TFAM (-/-) in mice. (Hamanaka et al 2016 DOI:https://doi.org/10.1016/j.celrep.2016.03.044).
- c) Respiratory chain inhibitors e.g. CI inhibitor BAY 87-2243, are inactive in RCC (Ellinghaus et al 2013, doi: 10.1002/cam4.112), and show reduced efficacy (growth inhibition) in hypoxia (Thomas et al 2019 doi.org/10.1186/s40170-019-0194-y).

VHL was not identified in either hypoxia or galactose screens (our screen, and Arroyo et al 2016 or Jain et al 2020 screens), unlike the antimycin A screen (Jain et al 2016). While both antimycin A and hypoxia treatment leads to reduced respiratory chain activity and ATP, the former blocks HIF and the latter induces HIF. Furthermore, hypoxia (our study) is not toxic, while antimycin A (Jain et al 2016) is toxic to cells. Thus collectively, these differences in VHL are not surprising, as complex III inhibition by antimycin A treatment is clearly a different condition to hypoxia or galactose treatment.

We have added more detailed discussion regarding the previous screens and their findings in the context of our study and findings (In 123-130, 199-202, 307-376).

Reviewer #2 (Remarks to the Author):

In this study, the essentiality of all genes for the cellular response to hypoxia was surveyed using a CRISPR screen, with an enrichment of nuclear encoded mitochondrial genes reported as essential. While the results are interesting, novelty is a problem. There are also some potential statistical issues, and places where clarity could be enhanced.

Major points:

(1) The study appears to be very similar to the recently published paper from Isha Jain & Vamsi Mootha (ref' #12). It is therefore incumbent on the authors to delineate what is new and/or different here. It is not enough to simply refer to this paper deep in the results with a throwaway "consistent with a recent study".

The major difference from the previous screen Jain et al (Cell 2020), is that our screen was performed in normoxia, hypoxia and galactose conditions in one study. Jain et al 2020 used a non-adherent immortalized chronic myelogenous leukemic cell line - K562. This K562 cell line was also used in a complex III respiratory inhibitor CRISPR screen from the Mootha group described in Jain et al 2016 Science 352(6281): 54–61 (see below). It is not clear from the authors why the K562 cell line was chosen for use in these 3 screens (Arroyo et al 2016. Cell Metab; Jain et al 2016 Science; Jain et al 2020 Cell). But it is particularly note-worthy to point out that the K562 cell line used these 3 studies from the Mootha group (Arroyo et al 2016. Cell Metab; Jain et al 2016 Science; Jain et al 2020 Cell) has been shown to carry mutations in several genes that might perhaps unknowingly affect metabolism and/or the HIF pathways (see https://www.cbioportal.org/patient?sampleId=K562_HAEMATOPOIETIC_AND_LYMPHOID_TISSUE&studyId=ccl_broad_2019). For example, K562 cells carry a mutation in EPAS1/HIF-2A, which is predicted to generate a shortened protein lacking the oxygen dependent domain which would lead to constitutively stabilised HIF-2 α in normoxia, and could have impacted the results of these 3 previous screens. As outlined above, we chose to use U2OS cells for our study for the reasons outlined above and because they carry no known mutations in the canonical HIF pathway genes.

The cells chosen for our screen (U2OS-HRE-luc) were generated from parental U2OS osteosarcoma cells, and originally described by us in a high throughput primary screening context before (Chau et al 2005 Can Res 65(11):4918-28). These cells contain a hypoxia/HRE luciferase reporter, and were used to generate the Cas9 expressing cells for our screen. Importantly, these cells have both a normal HIF response in hypoxia (new Supplementary Fig. 1c), and show enhanced OXPHOS when cultured in galactose (Supplementary Fig. 6c,d). Our screen analysed viable, adherent cells surviving after 5 days in culture conditions, which were harvested and counted prior to PCR/sequencing. These culture conditions are analogous to our small molecule inhibitor screen (Chau et al 2005 Can Res 65(11):4918-28), and subsequent screens which have which successfully identified novel (mitochondrial and non-mitochondrial) inhibitors of hypoxia signalling and tumour cell growth.

Since the results of the other paper are publicly available in the supplemental files, it would be useful to do a direct comparison across all the genes reported here and in the other paper, and provide some figures on the degree of overlap (e.g. Venn diagrams). This would enhance the data set in the current paper. The discussion could then be focused on what came out differently in the two studies.

Thank you. We have expanded our discussion and comparison with the recent study by Jain et al (Cell 2020) hypoxia screen (ln 125-128, 199, 307 onwards). Briefly, Jain et al 2020 study provides 'essential' gene data only on 21% versus 1% O₂ (Fig 1), as we do here. However, drilling down there is little 'gene by gene' agreement between the two studies, though one of the major findings – that mitochondrial genes (related to OXPHOS) are selectively essential in normoxia, is the same. However, we do not identify peroxisomal and lipid metabolism genes as a top pathway depleted (essential) in our screen (hypoxia (1 % O₂) compared to normoxia), unlike the Jain 2020 study (Fig 4). It is very important to point out that in the Jain et al 2020 study, the authors could not recapitulate the peroxisome/lipid finding from their screen when they validated these findings in the screening cell line (K562), nor could they recapitulate the peroxisome/lipid finding in other cells lines (e.g. HeLa, MCH58). In fact, they were only able to recapitulate the peroxisome/lipid finding in one cell line (HEK293Ts). In contrast to the Jain et al 2020 hypoxia study, in our study, all the top genes described from our screen were shown to recapitulate in the same cell line as the screen (U2OS), along with several other cell lines (HeLa, MCF7). Thus, completely validating the results from our screen. Another unexpected finding in the Jain et al 2020 hypoxia study, unlike our study, was that there was no change in translation-related genes, and yet it is well known that translation is suppressed in hypoxia in many cell lines. Conversely, in our screen we identify that loss of genes related to translation and cytoskeleton dynamics improve cell fitness in both hypoxia and galactose. These novel observations were not identified by either study by the Mootha lab (Jain et al Cell Metabolism 2016 (galactose screen) or Jain et al Cell 2020 (hypoxia screen)). These differences are likely a reflection of the strong cell specificity of gene essentiality under different conditions. Screens of this kind require considerable investment of resources, and so it is important that multiple studies using different cell models from different groups are available for the scientific community to clarify common and cell type specific gene essentiality. Indeed, one of the strengths of our study is the use of publicly available datasets from the Broad Institute which allow us to identify common essential genes under standard culture conditions (normoxia, glucose) across a large number of different cell types, and compare these with our screen.

We have added more detailed discussion regarding this previous screen and their findings in the context of our study and our findings (ln 123-130, 199-202, 307-376).

(2) The essentiality data set covered 1137 of the 1146 genes in MitoCarta. What were the missing ones? One presumes these missing 9 genes were ones that could not be CRISPR'd in the original screen? This needs to be stated.

Thank you. This is indeed the case, and we have clarified this in the text (ln 99).

(3) The paper in a number of places uses qualitative terms or incorrect language which does not enhance the narrative. e.g., line 67 "a large number of" (define large). Line 72 "we have provided" - since the screen was not actually conducted by you this seems to be a bit of a stretch.

Thank you. We have edited the text throughout to improve our use of terminology (e.g. ln 105, 115).

(4) It is stated that the experiments were performed in triplicate, but because these are cell culture experiments, it is not clear if these are simply 3 technical replicates (i.e. 3 flasks of cells from the same batch, or maybe different passages of the same cells), or they are true biological replicates (e.g., 3 completely independent stocks of cells prepared at different times). Either way, an N of 3 is at the very lowest end of the scale for statistical rigor. A power analysis should be included, to determine what effect size the study was capable of detecting with this number of replicates.

Thank you. We have edited the text to clarify the number of biological replicates, included an expanded method (ln 478-520), and included a new figure Supplementary Fig. 1a outlining the CRISPR screening approach we took here. To clarify, we carried out three biological replicates (three independent library infections with normoxia incubations, three independent library infections with hypoxia incubations, and three independent library infections with glucose-free (galactose) incubations. The triplicate replicates are considered independent biological replicates – separate flasks of cells are each individually transduced with lentivirus separately, and subsequently undergo antibiotic selection and passage as separate flasks throughout the complete duration of the experiment.

Our experimental set up matches the number of replicates described in the recent Jain et al study (2020), and constitutes one more replicate than the Arroyo et al study (2016). A power calculation was not included in either of these publications.

In reality today most screens are actually run as biological duplicates given recent studies that have compared the ability to detect depletion of known essential genes using 1 vs 2 vs 3 replicates. There is little improvement when moving from 2 to 3 samples if modern 2nd generation vector backbones are used (as in the library used in this paper). Also, importantly for statistical considerations, at least 5 gRNA per gene in the library – (see Figure 5 in Ong et al, Sci Rep. 2017; 7: 7384 -DOI: 10.1038/s41598-017-07827-z) should be considered. In our screen, each gene is targeted by at least 5 independent sgRNAs in each biological replicate, increasing the statistical power of each experiment.

Collectively, for these reasons, we are confident that the triplicate samples used in our experimental design is a robust approach for gene silencing at genome scale.

(5) Line 98. "Good agreement" with the Achilles Project data is not really supported by Suppl. Fig. 1d. The Venn diagram shows that of 264 mito genes in the Achilles set, only 109 (41%) were also found here. Likewise for all genes, only 49% were overlapped. The Venn diagrams should at least be drawn to scale, so that the degree of overlap is accurately represented.

Thank you. In this respect, "good agreement" was intended to refer to the identity of the most significantly depleted gene sets in both the Achilles data set and our own screens. We have modified the supplementary figure to include gene set overrepresentation analysis using the Achilles data set (Supplementary Fig. 2h) and have edited the text accordingly (ln 155-159).

Minor points:

(1) The rationale for selection of U2OS cells needs to be stated and justified.

Thank you, we have included a justification for the use of U2OS cells (ln 130-148).

(2) The Seahorse OCR data in Suppl. Fig. 4 are not normalized to anything. The method states the data were normalized to cell count, but the y-axis label does not include this information (i.e. pmols/min/n-cells).

Thank you, we have modified the y-axis label to take this into account (see revised Supplementary Figure 6c).

Reviewer #3 (Remarks to the Author):

In the manuscript by Thomas et al, the authors defined mitochondrially essential genes encoded by nuclear genes and performed two genome wide CRISPR/Cas genetic screens under conditions of metabolic stress. The first screen was performed in U2OS cells under hypoxia and the second performed with galactose to promote the use of OXPHOS for ATP production. The hypoxia screen identified SDHC and the galactose screen ROS genes.

While the quality of the writing in the manuscript is good, the Introduction and Discussion are extremely underdeveloped and it lacks an abstract to summarize the study. The formatting of this manuscript is far off what would be expected at Communications Biology. Once the authors address some of the issues mentioned above, I would be happy to review the manuscript.

Thank you. As the paper was a transfer into Nature Communications Biology, it was in an alternative text format for review (as permitted by the Editor). We have expanded the text to improve both the Introduction (ln 40-75) and Discussion (ln 311-381), and have included an Abstract (ln 26-39).

In the meantime:

1. SDHC is a critical gene for both the TCA cycle and OXPHOS, could the authors elaborate on why this was a top hit in the hypoxia screen?

Thank you. We apologise if this was not clear. SDHC sgRNA enrichment – along with numerous other mitochondrial genes involved in OXPHOS – is a top hit for hypoxia due to the improvement in fitness that loss of mitochondrial activity confers on U2OS cells in hypoxia. Indeed, loss of mitochondrial gene expression is a well-characterised physiological response to hypoxia exposure, as we confirm here. Our results with respect to mitochondrial gene essentiality in hypoxia are also consistent with a separate hypoxia CRISPR screen recently published (Jain et al 2020). The improvement in cell fitness is likely due to the bioenergetic benefit of diverting glucose-derived pyruvate from mitochondrial OXPHOS to ATP production via fermentation to lactate.

We have clarified this in the text (ln 244-250, 311-381) and revised Fig 3.

2. Is SDHC essential across multiple cancer types of where hypoxia plays a key role? Or is its role restricted within sarcoma?

Thank you, and we apologise if this was not clear. SDHC is a common essential gene in the Achilles data set, so yes, it is essential across many cancer types (Fig 1c). We have clarified this information in the text (ln 231-234).

3. Could the authors comment on the inter-relatedness of the two screens?

Thank you. We apologise if this was not clear. We have identified a list of genes that are enriched in both hypoxia and galactose treatment. Please see revised text ln 278-281.

REVIEWERS' COMMENTS:

Reviewer #1 (Remarks to the Author):

Well done!
I am satisfied.

Reviewer #2 (Remarks to the Author):

The authors have addressed all of my previous concerns, and the revisions to the manuscript result in a much-improved paper. I see no further obstacles to publication of this work.